# Competency-Based Education as Curriculum and Assessment for Integrative Learning

**Kayla M. Marcotte *** and **Larry D. Gruppen**

Department of Learning Health Sciences, University of Michigan Medical School, Ann Arbor, MI 48109, USA; lgruppen@umich.edu

\* Correspondence: kaymar@med.umich.edu

**Abstract:** Integrative learning and competency-based education are both evolving as major themes in education. Integrative learning emphasizes that knowledge from different domains and contexts are brought together to enhance the learner's experience. The emphasis on integrated learning has sparked the development of integrative curriculum, which methodically brings knowledge and skills together in ways that reinforce learning. Competency-based medical education (CBME) is an educational method that assumes integrative learning by relying on defined competencies for learners to master during their education. CBME is an illustration of both integrated learning and integrated curricula. In CBME, learners progress through their program by demonstrating acquisition of competencies, which are often integrative statements in themselves. In integrative learning, the question of how to assess a learner's progress through their program remains a challenge. Entrustable Professional Activities (EPAs) are one tool utilized to assess learning in CBME. EPAs are defined, observable tasks that learners should be able to demonstrate upon entering their profession. Understanding EPAs and how they are used in CBME may provide a framework for assessing integrative learning in diverse educational contexts.

**Keywords:** competency-based medical education; integrative learning; integrative curriculum; assessment

## 1. Introduction

Integrative learning and competency-based education are educational methods whose implementation is expanding. In this paper, we explore themes related to integrated learning and competency-based education. We propose that competency-based assessment techniques may provide useful guidance in developing assessments for integrative learning and curricula.

Integration has become a major theme in education (Figure 1). It is an educational method for combining knowledge and skills from multiple academic disciplines [1]. Integrated learning is pursued through integrative curricula; curricula that methodically partner disciplines [2–4]. An integrative curriculum focuses on content organization, the systematic building and revisiting of topics as the learner progresses through the curriculum. Several types of integrative curricula have been established: horizontal integration, vertical integration, and spiral integration [3,5].

Integrated learning is designed to strengthen a learner's grasp and application of knowledge in diverse settings as well as with diverse problems. Learners' bases of knowledge and skills are expanded by revisiting topics and connecting them to other disciplines and contexts. This method of learning enhances learners' abilities to retain and apply information in an array of different settings.

Integrative curricula are the means for initiating and supporting integrated learning. However, the ability to authentically assess integrated learning remains a challenge for educators. As learners' knowledge and skills grow, assessing their abilities can be difficult, especially when assessing learning in diverse contexts.

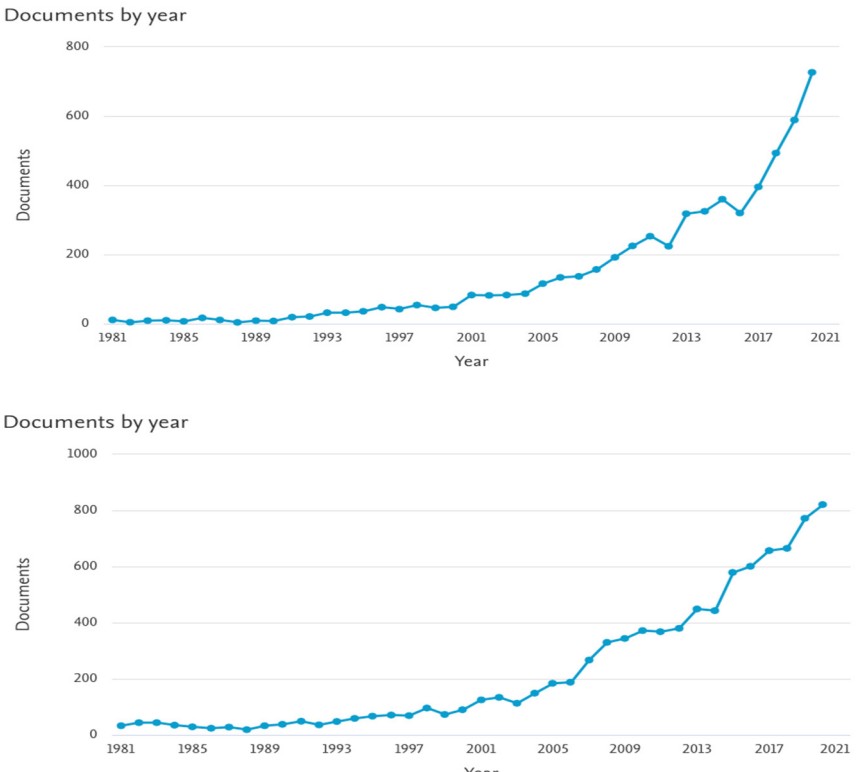

**Figure 1.** Number of documents published per year related to "integrated learning" (**top**) and "competency-based medical education" (**bottom**). Note that the vertical axes are on different scales.

Competency-based education is another educational method introduced in recent years that is well-represented within medical training (Figure 1). Competency-based medical education (CBME) focuses on learners achieving competencies as markers of their educational progress [6–8]. These competencies are constructed to demonstrate the knowledge and skills that learners have acquired through their training [8,9]. CBME focuses on individual, personalized learning that occurs at a pace set by the learner and consists of learning in the context of clinical practice, emphasizing formative assessments over summative ones. Integrative learning is at the core of competency-based education, as CBME is built upon integrating knowledge and skills as learners move through the curriculum.

Despite increases in research on both of these methods, few studies explore both integrated learning and CBME together (Figure 2). Integrated learning and integrative curricula are well developed within education, but accurately assessing integrated learning is still an issue. CBME has developed assessment techniques to evaluate how learners acquire and apply knowledge and skills in different settings. One specific manifestation of how integrated learning is assessed within CBME is through Entrustable Professional Activities (EPAs). EPAs are specific, measurable behaviors that build on the integrated learning of knowledge and skills in medical education [8,10–13]. In medicine, EPAs illustrate how integrated learning can be assessed as learners move through a curriculum. The parallels between integrative learning and CBME suggest that examining EPAs may offer a structure for developing assessment techniques that are applicable to integrated learning and curricula.

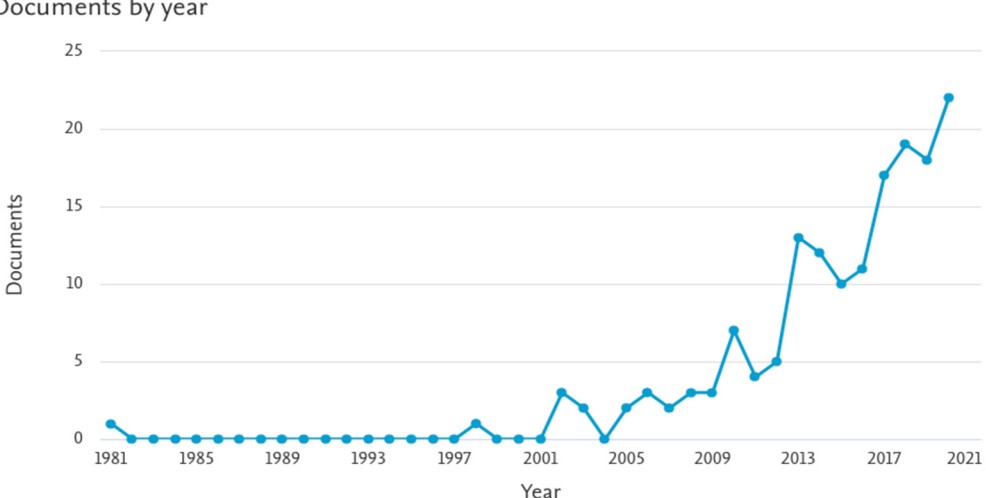

**Figure 2.** Number of documents published per year related to both "integrated learning" and "competency-based medical education".

## 2. Integrative Learning and Curricula

Harden defines integration is "an organization of teaching matter to interrelate or unify subjects frequently taught in separate academic courses or departments" [1]. Integrative learning occurs when knowledge from different subjects or disciplines is connected by the learner to enhance the learner's overall understanding.

All learning is integrative to some extent, as it integrates new experiences into an existing base of knowledge and skills. One way to promote integrative learning is through integrative curricula, where subjects and concepts are methodically connected and repeated in order to draw connections and enhance learning overall. Integrative learning can be enhanced by an integrative curriculum, which purposefully repeats and connects information in a methodical way to enhance learning. Since integrative learning is usually individualistic and unpredictable, faculty develop integrative curriculum to provide guidance for learners. This occurs by developing integrative curricula that explicitly and intentionally incorporate specific principles, skills, and knowledge configurations to achieve integrated learning.

Integrative curricula can take many forms. Horizontal integration requires integration across disciplines but across a finite period of time [3,5] Vertical integration includes integration of disciplines across time [3]. Spiral integration constitutes a combination of horizontal integration and vertical integration, where there is integration between disciplines as they are learned and across time as disciplines are revisited [3].

As with integrative learning, one could argue that all curricula are inherently integrative, but in the present context, we adopt Brauer and Ferguson's definition of the integrative medical curricula as "*a fully synchronous, trans-disciplinary delivery of information between the foundational sciences and the applied sciences throughout all years of a medical school curriculum*" [3]. In an integrative medical curriculum, the foundational sciences and clinical sciences are learned together and repeated throughout training. In this setting, integrative learning occurs by connecting the foundational sciences to the clinical sciences and vice versa, leading to a deeper understanding of each and a stronger foundation of knowledge to build on. Integrative learning in medicine also includes building clinical skills based on the learner's foundation of knowledge, which is another key aspect of medical training.

## 3. Competency-Based Medical Education Assumes Integrative Learning

Competency is, by definition, an integrated construct. Boyatzis defined competency as a "capability or ability", "a set of related but different sets of behavior organized around an underlying construct 'intent'" [6]. In medical education, Carraccio defines competency as a complex set of behaviors built on the components of knowledge, skills, attitudes, and "competence" as personal ability [9]. We use Frank et al.'s definition of competency: "An observable ability of a health professional, integrating multiple components such as knowledge, skills, values, and attitudes. Since competencies are observable, they can be measured and assessed to ensure their acquisition. Competencies can be assembled like building blocks to facilitate progressive development" [7].

Competency-based medical education (CBME) is one manifestation of integrated learning but one that is seldom discussed in the integrative learning literature. CBME is a curricular construct that allows learners to move through their medical training by demonstrating the achievement of different competencies. The design of CBME allows individual learners to move through curricula at individual paces and makes space for the flexibility in how and when different competencies are achieved. Learning and assessment of competencies often occurs within the context of practice.

Competences are inherently integrated in terms of connecting foundational sciences and clinical sciences. Integration is necessary in the set of behaviors required to demonstrate competence, but also in the knowledge that underlies the competencies. In medicine, competencies are often written as integrative statements (Table 1).

**Table 1.** Competency examples as integrative statements.

| Context | Competency Examples |
|---|---|
| Entering Medical School | Scientific Inquiry: Applies knowledge of the scientific process to integrate and synthesize information, solve problems, and formulate research questions and hypotheses; is facile in the language of the sciences and uses it to participate in the discourse of science and explain how scientific knowledge is discovered and validated [14]. Living Systems: Applies knowledge and skill in the natural sciences to solve problems related to molecular and macro systems including biomolecules, molecules, cells, and organs [14]. Critical Thinking: Uses logic and reasoning to identify the strengths and weaknesses of alternative solutions, conclusions, or approaches to problems [14]. |
| Clinical Sciences | Gather complete and focused histories from patients, families, and electronic health records in an organized manner, appropriate to the clinical situation and the individual, interpersonal, and structural factors that impact health [15]. Establish and maintain knowledge necessary for the preventive care, diagnosis, treatment, and management of medical problems [15]. Apply established and emerging evidence to diagnostic decision-making and clinical problem-solving [16]. Demonstrate and apply relevant scientific knowledge of the mechanisms of disease and of the consequences of abnormal physiology and anatomy [17]. Explain the molecular, biochemical, and cellular mechanisms that are important in maintaining the body's homeostasis [17]. |
| Foundational Sciences | Graduating students should demonstrate a broad base of established and evolving knowledge within a chosen discipline and detailed knowledge of a specific research area. Examples of discipline-specific conceptual knowledge include: proficiency in analytical approaches to defining scientific questions; broad-based knowledge acquisition; detailed knowledge of a specific research area [18]. Identify important, novel questions and critically design and execute experiments to address these questions [19]. |

## 4. Competency-Based Medical Education as Integrative Curriculum

Competency-based medical education does not require a specific educational method in order to achieve learning. In fact, CBME is designed to be flexible to enable the learner to achieve different competencies at their own pace. Thus, CBME can incorporate lectures, case-based learning, longitudinal courses, and many other educational methods.

CBME is particularly conducive to spiral curricula because competencies are designed to be achieved and revisited across time as learners progress through their program. For example, medical training may emphasize foundational sciences across subjects early in the curricula (horizontal integration), but as learners move towards clinical sciences they build on their knowledge of foundational sciences and revisit these subjects (vertical integration). This process repeats as learners build clinical knowledge and develop clinical skills (spiral integration).

A major component of CBME that promotes integrative learning is that learning often occurs in the context of clinical practice. The clinical learning environment promotes the cycled learning described in an integrative spiral curriculum because learners draw on their knowledge of foundational and clinical sciences to make clinical decisions. Learners participate in the regular cadence of seeing patients, finding and applying scientific knowledge, and revisiting patients. This cycle promotes integrative learning by requiring learners to consistently draw connections between the knowledge and skills they have developed over time.

## 5. Entrustable Professional Activities as an Assessment Tool for Integrated Learning

Developing meaningful and trustworthy methods to assess learning is paramount in education, but it is challenging to design assessments of integrated learning due to the multimodal nature of learning that must occur. Entrustable Professional Activities (EPAs) are one way CBME operationalizes integrated learning and assessment of outcomes. The American Association of Medical Colleges (AAMC) defines EPAs as "units of professional practice, defined as tasks or responsibilities that trainees are entrusted to perform unsupervised once they have attained sufficient specific competence. EPAs are independently executable, observable, and measurable in their process and outcome, and, therefore, suitable for entrustment decisions" [13,20]. They require specific knowledge, skills, and attitudes essential to the profession and they are observable and measurable [11]. EPAs are utilized in undergraduate and graduate medical education, as well as other training programs (Table 2).

**Table 2.** Examples of Entrustable Professional Activities (EPAs).

| EPA Examples | Source |
| --- | --- |
| Gather a history and perform a physical examination | Core Entrustable Professional Activities for Entering Residency [21]. |
| Form clinical questions and retrieve Evidence to advance patient care | Core Entrustable Professional Activities for Entering Residency [21]. |
| Resuscitate, stabilize, and care for unstable or critically ill patients | Internal Medicine End of Training EPAs [22]. |
| Provide general internal medicine consultation to non-medical specialties. | Internal Medicine End of Training EPAs [22]. |
| Design and implement a curricular intervention | MHPE [23]. |
| Create a learning plan | MHPE [23]. |

In medicine, an EPA is created by taking educational outcomes and translating them into an essential, observable, and measurable activity that a professional should be able to perform [8,10]. EPAs that require the consolidation of knowledge across domains, and contexts allow learners to show that they have achieved integrated learning when they

satisfactorily complete the EPA [13]. In this way, EPAs serve as a curricular tool and assessment tool for integrative learning [10,13,24].

Although EPAs provide opportunities for holistic performance, the quantification of this performance is complex and usually requires human judgment. A consequence of this is that EPAs often require observations by a skilled and knowledgeable human rather than completion of a test or similar technology. The dependence of EPAs on expert observer judgment introduces the potential for biases in the judgments. This is a significant challenge to assessing integrative learning because there may not be many faculty who, themselves, have an adequately integrated base of expertise.

EPAs may be assessed for the process or for the product of the activity. For example, an EPA may reveal the correct integration of knowledge and skills by an appropriate differential diagnosis (the product of the EPA) or by probing the cognitive processes of how evidence was gathered and weighted to arrive at this differential. Assessing the product is often simpler than assessing the process because products tend to be enduring whereas evidence or the process is often fleeting. Processes usually reflect a variety of internal, cognitive activities that are difficult to observe or capture. Products are usually concrete.

Complex EPAs in medicine are often team efforts, such as a successful emergency intubation. The individual's success in completing that EPA depends greatly on the performance of others and on the environment of the activity. It is challenging to attribute an outcome to specific individuals.

The complexity of the task identified in a given EPA may require subdivision into constituent tasks, depending on the level of experience of the learner and the goals of instruction or assessment. The EPA describes an essential activity for a medical professional (gathering a history and performing a physical exam). This activity is complex and integrates a considerable amount of knowledge acquired at varied times in classrooms and clinics. This EPA consists of key functions and related competencies, such as demonstrating patient-centered interviewing skills and demonstrating clinical reasoning. The learner behaviors that evidence the quality of performance on the EPA are ordered in a sequence that reflects inadequate, developing, or expected levels of performance.

It will be apparent that, on one hand, using EPAs as an assessment tool for integrative learning is much more complex than the familiar multiple choice tests. EPAs also require considerably more development time and effort, to say nothing of the faculty expertise demanded. On the other hand, EPAs offer a level of authenticity and real-world evidence that is impossible with standardized tests. They require higher levels of learning ("application of knowledge" rather than simply "recognition") and collaboration among foundational science departments and faculty. They are sufficiently complex, such that no single faculty member can effectively design one or develop the assessment components of the EPA. Instead, developing EPAs is generally an institutional effort, as is the assessment process and the decisions made about learner competence.

## 6. Conclusions

Integrative learning reflects the reality of a highly interconnected world in which knowledge cannot be isolated into specific silos or applied in prescribed ways. The complexity of integrative learning requires innovation in curricular design, but it also requires innovative learner assessment. Assessment needs to be as interdisciplinary and synthetic as the underlying learning. Competency-based education provides one useful model for both curriculum and assessment.

Competencies are, by nature, integrative and provide a framework for organizing and learning relevant knowledge, skills, and values in appropriate contexts. However, perhaps even more useful are the implications for assessment. Traditional, disciplinary tests are poor indicators of the complex behaviors and applications of integrative learning. Instead, assessments must reflect the nature of integrative learning by focusing on the application of complex, richly connected knowledge in familiar and novel situations. Entrustable Professional Activities provide such an assessment framework.

Such assessment is a major commitment of and investment by the institution, which is where much of this work will need to be done; it is beyond the capacity of most individual faculty. However, its value lies in making educational decisions that are more predictive of real-world performance after training. It emphasizes assessment for learning rather than only of learning. By understanding and adapting assessment techniques for integrative learning, educators and learners can enhance the overall educational experience offered by integrative learning.

**Author Contributions:** Conceptualization, K.M.M. and L.D.G.; methodology, K.M.M. and L.D.G.; investigation, K.M.M. and L.D.G.; resources, K.M.M. and L.D.G.; writing–original draft preparation, K.M.M. and L.D.G.; writing–review and editing, K.M.M. and L.D.G.; visualization, K.M.M. and L.D.G.; supervision, K.M.M. and L.D.G.; project administration, K.M.M. and L.D.G. All authors have read and agreed to the published version of the manuscript.

**Funding:** This research received no external funding.

**Institutional Review Board Statement:** Not applicable.

**Informed Consent Statement:** Not applicable.

**Data Availability Statement:** Not applicable.

**Conflicts of Interest:** The authors declare no conflict of interest.

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
