# Peer review of "Competency-Based Education as Curriculum and Assessment for Integrative Learning"

_education, doi:10.3390/educsci12040267_

Round 1
Reviewer 1 Report
The manuscript consists of total 9 pages, including 3 figures, 2 tables and the list of total 24 literature references. The review article discusses modern idea of holistic education and evaluating its effects, including integrative learning and curricula, compentency-based medical education, entrustable professional activities, thus it fits into the scope of topics raised by the Journal. The article is logically organized and written in very good quality English, which makes it an excellent and pleasurable read. The educational value of this review is unquestionable. It is of especially high value that the Authors presented the ideas from both sides, not only stressing their strengths but also pointing at some of their weaknesses and limitations. It is not the main role of the reviewer to come into a polemic with the ideas presented by the Authors in the reviewed text but I feel that it is my duty of as a quite experienced academic teacher to point at some observation the Authors may want to utilize somehow in the the conclusions. Namely, it is an observable fact that the education seems to have made a circle and what we actually can observe nowadays is that - after decades of striving for standardization, unification and comparability of both the processes of teaching and learning and of the processes of evaluation of teaching and learning and their effects - it is coming back to its starting point, which (if we unwrap it from all the scientific jargon and drag it from behind strict but complicated definitions, punctuated lists and tables) is the highly individualized, unique and difficult to evaluate objectively or compare against other people relation of the type master-apprentice, in which the apprentices' learning effect is (not) good enough as long as the master does (not) consider it as such. That idea of total individualism in education may be tempting (especially to students) but in the end of the day we are all left on the verge of educational nihilism: with a wide collection of (to some undetermined extent) educated people (confirmed as such based on the highly subjective say so of their teachers) who cannot be in a fast and easy way compared against each other, made into a team composed of members of proven basic compatibility, or just ranked in order to evaluate how likely they actually are to perform good (or bad) in doing this or that job or where they shall go to study further or what (concrete) they shall make up in case they need to switch between schools or educational programs. All this requires assessments, which again are defined in complex, general and thus unclear ways, often (again) to the extent inviting sheer subjectivity in the evaluators. The whole educational environment becomes liquid, blurred, if not muddy, unsteady and unpredictable, thus perceived as unsafe or sometimes even hostile - both for the students and teachers. I want to stress again, the above general reflections do not undermine the unquestionable value and high scientific quality of the reviewed manuscript.
The Abstract does not mirror the structure of the main text but provides enough information for the Readers to get engaged with the text. What I am missing here is some hint of the Authors' own position or opinion about the raised topic.
The Introduction provides enough background information for the main topic of the manuscript.
There is no Material and method section in the text and the Readers could benefit from a short information on how the Authors have selected the sources they referred to in the article.
There is no Results section with a separate header, which shall be added to the text for the sake of clarity of composition, before the header No 2. The review is clear and comprehensive, adequately illustrated and augmented with examples, which additionally add to the high value of the text.
There is no Discussion section either, which is acceptable in a review article.
The Conclusions are logically consistent with the previous parts of the text.
The Referred literature is numerous, recent enough and relevant to the raised topic.
Author Response
Dear Reviewer,
Thank you for your helpful comments for how we can improve our manuscript. We have made the following changes to our paper following your suggestions:
- Restructured the abstract to better mirror the paper’s structure and added more of our stance on the subject to the abstract.
- Added a Methods section to discuss our sources
- Added “Results” to clarify the structure of the paper, with the subsequent sections listed as subheadings.
We hope these changes are satisfactory and thank you again for taking the time to review our manuscript.
Reviewer 2 Report
This is well written review of the importance of integrative learning and competency based education. Including a review of EPAs. This is an important hot topic in medical education.
Author Response
Dear Reviewer,
Thank you for your comments on our manuscript. We appreciate you taking the time to review our work.